# Resistance to Targeted Inhibitors of the PI3K/AKT/mTOR Pathway in Advanced Oestrogen-Receptor-Positive Breast Cancer

**DOI:** 10.3390/cancers16122259

**Published:** 2024-06-18

**Authors:** Iseult M. Browne, Alicia F. C. Okines

**Affiliations:** Breast Unit, The Royal Marsden NHS Foundation Trust, London SW3 6JJ, UK; iseult.browne@icr.ac.uk

**Keywords:** advanced breast cancer, ER-positive breast cancer, PI3K inhibitors, AKT inhibitors, mTOR inhibitors, treatment resistance

## Abstract

**Simple Summary:**

The outcomes for patients with advanced oestrogen-receptor-positive (ER-positive) breast cancer have improved significantly over the last decade. Whereas treatment of endocrine-resistant disease was previously limited to conventional chemotherapy, we are now seeing increasing approvals for targeted agents suitable for combination with hormone therapies to potentially reverse the endocrine resistance for these patients in the early lines of treatment. The phosphoinositide-3-kinase (PI3K)/AKT/mammalian target of rapamycin (mTOR) pathway is frequently dysregulated in human cancers, and though many drugs targeting this pathway have been developed, a vast number of these compounds have failed in clinical trials, with both toxicity and resistance being major problems. In this article, we summarise the PI3K/AKT/mTOR pathway, review the mechanisms of resistance to agents that target this pathway, and discuss future potential strategies to overcome this resistance.

**Abstract:**

The PI3K/AKT/mTOR signalling pathway is one of the most frequently activated pathways in breast cancer and also plays a central role in the regulation of several physiologic functions. There are major efforts ongoing to exploit precision medicine by developing inhibitors that target the three kinases (PI3K, AKT, and mTOR). Although multiple compounds have been developed, at present, there are just three inhibitors approved to target this pathway in patients with advanced ER-positive, HER2-negative breast cancer: everolimus (mTOR inhibitor), alpelisib (PIK3CA inhibitor), and capivasertib (AKT inhibitor). Like most targeted cancer drugs, resistance poses a major problem in the clinical setting and is a factor that has frequently limited the overall efficacy of these agents. Drug resistance can be categorised into intrinsic or acquired resistance depending on the timeframe it has developed within. Whereas intrinsic resistance exists prior to a specific treatment, acquired resistance is induced by a therapy. The majority of patients with ER-positive, HER2-negative advanced breast cancer will likely be offered an inhibitor of the PI3K/AKT/mTOR pathway at some point in their cancer journey, with the options available depending on the approval criteria in place and the cancer’s mutation status. Within this large cohort of patients, it is likely that most will develop resistance at some point, which makes this an area of interest and an unmet need at present. Herein, we review the common mechanisms of resistance to agents that target the PI3K/AKT/mTOR signalling pathway, elaborate on current management approaches, and discuss ongoing clinical trials attempting to mitigate this significant issue. We highlight the need for additional studies into AKT1 inhibitor resistance in particular.

## 1. Introduction

The PI3K/AKT and mTOR signalling pathway plays a crucial role in many normal cellular processes [1], responding to hormones, nutrients, and growth factors. This pathway is one of the most frequently activated pathways in breast cancer [2], (Figure 1) evident in up to 70% of breast cancers, making it an attractive target for drug development [3]. Hyperactivation of this pathway frequently signifies the development of drug resistance, including resistance to endocrine therapies [4,5].

The most common dysregulation of this pathway occurs due to mutations in *PIK3CA,* which occur in all sub-types of breast cancer but are most common in ER-positive, HER2-negative cancers, present in up to 40% of cases. The majority of mutations are located at two hotspots on exons 9 and 20 [6]. This induces a hyperactivation in p110α. In an estimated 7% of advanced ER-positive breast cancers, the PI3K pathway is alternatively activated by a mutation in *AKT1* [7]. Approximately 80% of these *AKT1* mutations occur at E17K which leads to constitutively active PI3K signalling [2,8,9]. Loss of function *PTEN* mutations are observed in up to 18% of ER-positive breast cancer cases [10].

PI3K/AKT/mTOR pathway activation usually occurs due to growth factor-mediated induction of G-protein coupled receptors (GPCRs) or receptor tyrosine kinases (RTKs) [11,12], and alterations in certain receptors such as the epidermal growth factor receptor (EGFR), vascular endothelial growth factor receptor (VEGFR), and the insulin receptor (INSR) can lead to a significant level of treatment failure through the activation of the PI3K/AKT/mTOR pathway [13,14,15].

In ER-positive breast cancer, the PI3K/AKT/mTOR pathway can also be activated as a method of acquired resistance to long-term oestrogen deprivation [16]. Crosstalk exists between the oestrogen receptor and the PI3K/AKT/mTOR pathway, and PI3K inhibition triggers the activation of ER-dependent transcription machinery which leads to enhanced oestrogen receptor function [17].

Though this pathway is an attractive target for treatment, the narrow therapeutic window, which is limited by on-target normal tissue toxicity, is a major challenge. At the time of writing, there are three drugs approved to target this pathway for patients with ER-positive, HER2-negative advanced breast cancer leading to both improved outcomes and a reduction in chemotherapy use. Unfortunately, to date, several other compounds have failed in clinical trials. One of the most common reasons for this failure is a drug’s inability to downregulate this pathway to levels required for a response in the tumour. This is often mediated by the activation of feedback loops, which leads to the reactivation of the signalling pathway, a mechanism of intrinsic adaptive response to targeted therapy [18]. Compensatory parallel signalling pathways can also be activated, such as ER signalling in the case of the PI3K inhibition, despite co-administration with endocrine therapy [17]. Pathway reactivation due to compensatory mechanisms is particularly challenging for the PI3K/AKT/mTOR pathway, which has many physiologic feedback loops that can become induced on inhibition of the pathway. The development of acquired resistance can occur by many mechanisms which we will discuss further; in the case of p110α-specific *PIK3CA* inhibitors such as alpelisib, PTEN loss is one such way [19]. To overcome this resistance, ongoing trials of dual inhibitors are in place, and the role of triplet therapy is being evaluated.

## 2. Targeting of the PI3K/AKT/mTOR Pathway in Breast Cancer

### 2.1. mTOR Inhibitors

mTOR is a serine/threonine protein kinase found downstream of PI3K and AKT, comprising two different complexes, mTORC1 and mTORC2 [20]. Whereas mTORC1 composes mTOR, Raptor, PRAS40, mLST8, and deptor and regulates cell growth and metabolism, mTOR2 comprises mTOR, proctor, rictor, mSIN1, mLST8, and deptor and primarily controls cell proliferation and survival. There are multiple critical inputs into TORC1 signalling, including growth factors such as insulin, EGF, tumour necrosis factor (TNF), wingless type integration site family (WNT) ligands, and amino acids [21]. mTORC2 is less understood than mTORC1; however, it is a key part of this pathway as it directly phosphorylates AKT [22]. As such, mTORC1 and mTORC2 function downstream and upstream of AKT, respectively.

Analogues of the natural product rapamycin (rapalogs), an inhibitor of the mTORC1 complex, were among the first drugs developed to target this pathway in the treatment of cancer. There are two main categories of mTOR inhibitors: allosteric mTOR inhibitors and ATP-competitive catalytic mTOR inhibitors and three generations of mTOR inhibitors [23]. Rapalogs do not directly inhibit mTORC1 but bind to immunophilin FK506 Binding Protein-12 (FKBP12). This complex binds to the FKBP-rapamycin binding (FRB) domain of mTOR and allosterically inhibits phosphorylation of the substrate S6 ribosomal protein kinase (S6K1) [24,25]. It thereby inhibits cap-dependent translation and cell proliferation. Rapalogs result in incomplete inhibition of MTORC1 as they only weakly inhibit the phosphorylation of eukaryotic elongation factor 4E-binding protein (4EBP1) [26]. The FKBP12-rapamycin complex cannot bind to mTORC2 which limits the acute inhibition of mTORC2. Inhibition of mTORC1 alone risks activating a negative feedback loop that leads to the activation of mTORC2 and AKT, potentially leading to treatment resistance [27].

Everolimus is a first generation rapalog which has been approved by the Food and Drug Administration (FDA) and European Medicines Agency (EMA) since 2012 in combination with exemestane, based on the results from the BOLERO-2 trial, for patients with ER-positive, HER2-negative advanced breast cancer following progression on a non-steroidal aromatase inhibitor (letrozole or anastrazole) [28]. In BOLERO-2, median PFS was improved from 3.2 to 7.8 months (HR, 0.45; *p* < 0.0001) with the addition of everolimus to the steroidal aromatase inhibitor, exemestane. Although there was a numerical improvement in median overall survival (OS) from 26.6 to 31.0 months, this was not statistically significant (HR, 0.89; *p* = 0.14) [29]. Studies investigating everolimus in combination with chemotherapy fail to demonstrate benefit, and no biomarker to predict everolimus sensitivity or resistance has been identified [30,31,32].

Second-generation mTOR inhibitors (ATP-competitive inhibitors) inhibit the kinase domain of mTOR, blocking both mTORC1 and mTORC2, though, to date, there have been no approvals in the advanced breast cancer setting [33]. Due to the kinase similarities between mTOR and PI3K, some of these second-generation mTOR inhibitors also block PI3K leading to a degree of vertical pathway inhibition [34]. Vistusertib is a dual inhibitor of mTORC1 and mTORC2 which demonstrates good activity in preclinical breast cancer models [35]. The phase II MANTA trial evaluated vistusertib in combination with fulvestrant compared with fulvestrant alone or fulvestrant plus everolimus in postmenopausal women with ER–positive advanced breast cancer. This trial failed to demonstrate a benefit in adding vistusertib to fulvestrant; in fact, everolimus alone demonstrated a significantly longer PFS: median PFS 8 months with fulvestrant plus vistusertib vs. 12.3 months with fulvestrant plus everolimus (HR, 0.63; 95% CI, 0.45–0.90; *p* = 0·01) [36]. A third generation of mTOR inhibitors has been developed to overcome resistance mutations to rapalogs and kinase inhibitors of mTOR. This molecule is known as rapalink and contains rapamycin crosslinked with a kinase inhibitor of mTOR [37].

### 2.2. PIK3CA Inhibitors

There are three classes of PI3Ks (Class I, II, and III) that are grouped in accordance to their structure, binding partners, and substrate specificity [38]. Class I PI3Ks are divided into class IA and IB, the former being the most studied in breast cancer. Class IA PI3Ks consist of two subunits, a p85 regulatory subunit and a p110 catalytic subunit, with activation by RTKs, GPCRs, and certain oncogenes such as RAS [39,40]. There are three isoforms of the p110 catalytic subunit of Class IA PI3Ks (p110α, p110β, and p110δ) that are encoded by the genes *PIK3CA*, *PIK3CB*, and *PIK3CD*, respectively [39]. The PI3Ks catalyse the phosphorylation of phosphatidylinositol 4,5 bisphosphate (PIP2) to generate phosphatidylinositol 3,4,4-triphosphate (PIP3). PIP3 is the key second messenger driving downstream signalling cascades and leads to the phosphorylation and activation of AKT and PDK1 [40].

The first generation of PI3K inhibitors, known as pan-PI3K inhibitors, targets all four catalytic isoforms of class I PI3Ks (α, β, γ, and δ) and includes pictilisib, buparlisib, and copalinisib. These compounds show lack of sufficient selectivity to p110α, and patients experienced high levels of treatment-related adverse events in clinical trials, including significant psychiatric/mood disorders. These compounds, therefore, never reached routine clinical practice [41,42].

The development of Isoform-selective PI3K inhibitors has greatly reduced the toxicity experienced with pan-PI3K inhibitors [43]. Alpelisib is an α-selective PI3K inhibitor which inhibits p110α to a much greater degree than other isoforms [44]. Alpelisib was approved in combination with fulvestrant by the FDA and EMA in 2019 based on the results of the phase III SOLAR1 trial for postmenopausal women with ER-positive, HER2-negative, *PIK3CA*-mutated, advanced or metastatic breast cancer post aromatase inhibitor (AI) [43]. This trial showed an improved PFS from 5.7 to 11 months (HR, 0.65; 95% CI, 0.50–0.85; *p* < 0.001) with the addition of alpelisib to fulvestrant in patients with *PIK3CA* mutations. Supporting data from the BYLieve trial confirmed maintained efficacy of alpelisib in patients who received a CDK4/6 inhibitor with the AI, although the magnitude of benefit appeared abrogated [45].

Though isoform-selective PI3K inhibitors such as alpelisib have greatly improved toxicity profiles, discontinuation of treatment due to side-effects remains an issue, as evidenced by the 25% discontinuation rate in SOLAR-1. Alpelisib inhibits both mutant and wild-type PI3Kα, and as such, the toxicity from inhibition of wild-type PI3Kα, including hyperglycaemia and rash, limits the tolerable dose of this drug. Ongoing trials are evaluating the role of mutant specific inhibitors in ER-positive, HER2-negative breast cancer patients [46,47]. For example, LOXO-783 is a potent mutant selective allosteric *PI3Kα H1047R* inhibitor, which induced significant tumour regression in ER-positive, HER2-negative, *PI3Kα* H1047R-mutant breast cancer models, and is being evaluated in the Phase I trial PIKASSO-01 (NCT 05307705) [46].

### 2.3. AKT Inhibitors

AKT and PDK1 are closely linked to PI3K and are key effectors of downstream signalling. PIP3 binds to the PH region of AKT, recruiting it to the plasma membrane to allow PDK1 to phosphorylate at the Thr308 region in the kinase domain [48]. A second phosphorylation is required at the regulatory domain Ser473 for full activation of AKT [22]. AKT consists of three isoforms (AKT1, AKT2, AKT3) and is the major downstream target of PI3K, exerting a pivotal role in regulating cell proliferation, metabolism, and survival. AKT modulates a myriad of substrates including tuberous sclerosis complex 2 (TSC2), murine double minute 2 (MDM2), glycogen synthase kinase 3beta (GSK3beta), and the forkhead family of transcription factors (FoxO) [49].

To date, multiple allosteric and ATP-competitive AKT inhibitors have been explored in clinical trials. MK2206 is the most prominent allosteric inhibitor which has demonstrated promising antitumour effects in preclinical breast cancer models but has not progressed further to regulatory approval due to limited efficacy and dose-limiting toxicity in phase 2 trials [50,51]. Ipatasertib and capivasertib are the most studied ATP-competitive inhibitors of AKT.

Capivasertib inhibits all three AKT isoforms and was recently approved by the FDA in combination with fulvestrant for postmenopausal women with ER-positive, HER2-negative advanced breast cancer post progression on an aromatase inhibitor, with one or more biomarker alterations in *PIK3CA*, *AKT*, or *PTEN*. This approval was based on the results from the phase III CAPItello-291 trial. In this trial, the addition of capivasertib to fulvestrant improved median PFS in the overall population, though the result was driven primarily by the benefit in patients with activating *PIK3CA*, *AKT1* mutations, or *PTEN* truncating mutations and variants known to inhibit PTEN function. The median PFS in the overall study population was prolonged from 3.6 to 7.2 months (HR 0.60, 95% CI 0.51–0.71, *p* < 0.001) with capivasertib, but the greatest benefit was seen in patients with pathway-activating alterations, in whom median PFS improved from 3.1 to 7.3 months (HR, 0.50; 95% CI, 0.38–0.65; *p* < 0.001) with the addition of capivasertib [52]. These data are supported by the preceding phase II FAKTION trial [53], which reported a median PFS of 10.3 vs. 4.8 months (HR, 0.56; 95% CI, 0.38–0.81; *p* = 0.0023), and the median OS was 29.3 vs. 23.4 months (HR 0.66, 95% CI 0.45–0.97; *p* = 0.035) for capivasertib vs. placebo, respectively, in the intention-to-treat population. In a subgroup analysis of patients with defined alterations in *PIK3CA*, *AKT*, or *PTEN*, median PFS was 12.8 vs. 4.6 months (HR, 0.44; 95% CI, 0.26–0.72; *p* = 0.0014), and median OS was 38.9 vs. 20.0 months (HR, 0.46; 95% CI 0.27–0.79; *p* = 0.0047) for capivasertib vs. placebo, respectively. Statistically significant differences were not seen in PFS or OS in the pathway-non-altered subgroup in the updated analysis: median PFS 7.7 vs. 4.9 months (HR, 0.70; 95% CI, 0.40–1.25; *p* = 0.23) and median OS 26.0 vs. 25.2 months (HR 0.86; 95% CI, 0.49–1.52; *p* = 0.60) [54]. See Table 1 for a summary of the results of randomised phase II/III trials of agents in ER-positive, HER2-negative advanced breast cancer and Figure 2 for the site of action of each of the agents.

## 3. Resistance

The PI3K/AKT/mTOR pathway is involved in numerous essential physiological functions, limiting the ability to target this pathway. The pharmacological targeting of this pathway can lead to the induction of feedback loops, activation of compensatory parallel signalling pathways, and crosstalk with other signalling nodes, all of which can oppose pathway inhibition (Figure 3).

### 3.1. Receptor Tyrosine Kinase Activation and Negative Feedback Loops

The inhibition of the PI3K/AKT/mTOR pathway can lead to the induction of RTKs including IGFR-1, EGFR, HER2, and HER3 [27,56]. This RTK activation can lead to the induction of downstream pathways including the PI3K/AKT/mTOR and Ras/Raf/MEK/ERK pathways. This can limit the response to compounds that target the PI3K/AKT/mTOR pathway [57].

Pathway reactivation via RTKs is driven by the loss of the inhibitory effect AKT has on the transcription factor FOXO and mTORC1. When the PI3K/AKT/mTOR pathway is activated, AKT phosphorylates and attenuates FOXO by the sequestration of FOXO in the cytoplasm. This reduces the ability of FOXO to upregulate the transcription of RTKs that are tightly coupled to PI3K including HER3, IGFR1, and insulin receptor, in turn limiting the induction of RTKs by extracellular stimuli [58]. AKT activation also leads to activation of TORC1 and S6K which inhibits IRS1 expression, resulting in reduced signalling.

With the use of PI3K inhibitors, the phosphorylation of FOXO by AKT is suppressed, leading to the stimulation of RTKs and partial restoration of PIP3 activity. As PIP3 activity is maintained to a certain level, PI3K signalling cannot be fully suppressed, which leads to cell proliferation. This mechanism of RTK activation is mediated by the PI3K p110α isoform, though in PTEN-deficient tumours, the main isoform driving PI3K signalling is p110β [59].

The pharmacological targeting of AKT has also been demonstrated to induce the phosphorylation and expression of multiple RTKs, leading to reduced anticancer activity [60,61]. Inhibition of AKT induces a conserved set of RTKs including IGF-1R, IR, and HER3, in part due to mTORC1 inhibition and in part due to a FOXO-dependent activation of RTKs. Chandarlapaty et al. report that inhibition of AKT with an allosteric inhibitor induced HER3 expression with induction of HER3-HER2 heterodimers and a marked increase in HER3 phosphorylation. When HER3 is phosphorylated, it has a high affinity for PI3K, leading to its activation [60]. Further study of this potential resistance mechanism is warranted, including the impact of HER3 inhibition.

### 3.2. Loss or Inactivation of PTEN

Phosphatase and tensin homologue (*PTEN*) is an important tumour suppressor gene which functionally antagonises PI3K activity, modulating this pathway by dephosphorylating PIP3 to PIP2 [62]. Germline mutations in *PTEN* are rare but give rise to Cowden’s syndrome, a multisystem autosomal dominant condition associated with a high risk of cancers including breast cancer [63].

A clinically validated mechanism of acquired drug resistance to p110α-specific inhibitors, such as alpelisib, is the acquisition of loss of function *PTEN* mutations, which in turn leads to increased signalling through the PI3K p110β isoform [19,64]. The initial efficacy of p110α inhibition may be mitigated by rapid accumulation of PIP3 produced by the p110β isoform [59]. Schwartz et al. identified that the use of a p110β inhibitor for PTEN-null cells led to decreased signalling and tumour cell growth, where selective p110α inhibition had no effect [65]. The inhibition of PI3Kβ only transiently inhibited AKT/mTOR signalling due to the fact it relieves the negative feedback inhibition of RTKs and leads to a rebound in downstream signalling as previously discussed. Importantly, this rebound is suppressed with the combined inhibition of PI3Kα and PI3Kβ, leading to increased antitumour activity.

Loss of function *PTEN* mutations can also occur with de novo resistance, as seen in an analysis of plasma and tumour samples from the phase I trial (NCT01870505) of alpelisib [66]. This mechanism could be potentially overcome using an AKT inhibitor, but further research is required.

### 3.3. Acquired Amplification and Mutation of Specific Genes

Acquired amplification and/or mutation of *PIK3CA* or *PIK3CB* have been demonstrated to cause resistance to PI3K inhibitors [67,68]. Huw et al. reported preclinical modelling of acquired resistance to the pan-PIK3CA inhibitor, pictilisib, in a *PIK3CA*-mutant breast cancer cell line. Huw et al. identified that amplification of the mutant allele of *PIK3CA* resulted in substantial upregulation of PI3K signalling and resistance. It was also shown that the knockdown of the amplified *PIK3CA* mutant allele restored pathway sensitivity to PI3K inhibition at levels comparable to parental cells [67]. Whether or not *PIK3CA* or B amplification similarly confers resistance to AKT1 inhibitors is currently unknown.

Nakanishi et al. demonstrated that activating *PIK3CB* D1067Y/A/V mutations confers resistance to PI3K inhibitors due to hyperactivation of the PI3K pathway. These mutations may be acquired to maintain the p110β signalling in the presence of PI3K inhibitors [68].

Upregulation of AKT3 has been shown to confer resistance to the allosteric pan-AKT inhibitor MK-2206. Stottrup et al. demonstrated that AKT3 expression is markedly upregulated in AKT-inhibitor-resistant cells. This induction of AKT3 was regulated by the bromodomain and extra terminal domain proteins, and knockout of AKT3 but not AKT1 or AKT2 in resistant cells restored sensitivity to MK2206 [69]. This mechanism of resistance has not yet been reported after treatment with capivasertib.

### 3.4. Activation of Parallel Pathways

Parallel activation of various signalling pathways and specific kinases that feed into the PI3K/AKT/mTOR signalling pathway might also give rise to primary resistance or the emergence of early resistance to targeted therapies. For example, as shown by Castel et al., in breast cancer cells resistant to p110α inhibition, the PDK1-SGK axis can overcome AKT inhibition by activating mTORC1 signalling [70]. Whether combination with mTORC1 inhibition or switching therapy to an mTOR inhibitor could be successful will require further study.

PIM1 is a kinase which is overexpressed in multiple malignancies and has been proposed to confer resistance to PI3K inhibitors including alpelisib via maintaining downstream pathway activation in an AKT-independent manner [71]. PIM1 kinase functions to reduce cellular ROS levels by enhancing NRF2/ARE activity. PIM1 has been shown to regulate mTORC1 and eIF4B leading to translational control of NRF2 levels. PIM1 also reduces ROS production, in turn attenuating PI3K/AKT-inhibitor-induced cytotoxic effects. Song et al. report that treatment with PIM kinase inhibitors reverses this resistance and increases a tumour’s susceptibility to PI3K/AKT-targeted inhibitors [72]. The clinical development of PIM kinase inhibitors has been limited to haematological malignancies; thus, further preclinical data would be required to inform potential combination studies in breast cancer.

### 3.5. Insulin Signalling and PI3K Reactivation

One of the most common adverse events seen in clinical practice in patients on PI3K inhibitor treatment is the on-target effect of hyperglycaemia [43,73], occurring via inhibition of the intracellular response to insulin. They additionally promote glycogenolysis in the liver and downregulate AKT2’s ability to regulate GLUT transport in adipose tissue [74]. This can precipitate the medical emergency of diabetic ketoacidosis if untreated.

Growth factor stimulation of insulin-like growth factor 1 receptor (IGF1R) and insulin receptor (INSR) leads to the phosphorylation of insulin receptor substrate (IRS) adaptor molecules (usually IRS1) which subsequently leads to PI3K activation (Figure 3). IRS1-induced activation of PI3K occurs via binding to the two SH2 domains on the PI3K p85 subunit, which triggers the induction of the PI3K/AKT/mTOR signalling axis [75]. The PI3K p85 subunit forms a sequestration complex with IRS1 when in surplus to the PI3K p110 subunit, preventing IRS1-induced activation of the PI3K/AKT/mTOR pathway, leading to inhibition of insulin signalling [76]. This mechanism can result in the inhibition of insulin signalling through the preservation of PI3K p85 subunit basal inhibition on PI3K p110 subunit and, concurrently, could further lead to insulin resistance if expressed in excess. Upon induction, the PI3K p110α subunit is then capable of mediating intracellular response-insulin stimulation, triggering growth and glucose homeostasis in the majority of tissues [77].

The glucose transporter family (GLUTs) is primarily responsible for the uptake of glucose into cells, and constitutive activation of AKT signalling promotes glucose uptake via GLUT1 and GLUT4. Inhibition of AKT also induces insulin receptor expression [60]. It has been shown that an increase in insulin and blood glucose levels after treatment with PI3K inhibitors leads to activated PI3K signalling via glucose–insulin feedback, even in the presence of PI3K inhibitors [78]. In cell lines resistant to alpelisib, Leroy et al. report an increase in IGF1R, IRS1/IRS2, and p85 phosphorylation, which provides a rationale for combining inhibitors targeting p100α with IGF1R inhibitors in breast cancer patients with *PIK3CA* mutations [79], though clinical trials to date of such combinations have not been promising [80].

Insulin signalling is feedback regulated partly by an mTOR/S6K-dependent phosphorylation and downregulation of IRS1 [81]. Inhibition of mTORC1 with rapamycin relieves this feedback which activates insulin and IGF signalling, in turn activating PI3K and ERK signalling, reducing the therapeutic efficacy of the drug [27].

### 3.6. Endocrine-Mediated Resistance

The crosstalk between the PI3K/AKT/mTOR pathway and the oestrogen receptor (ER) is extensive and underlies the importance of targeting both pathways simultaneously to overcome drug resistance. Preclinical and clinical data support a negative feedback model in which these pathways antagonise each other (Figure 3). Adaptation to long-term oestrogen inhibition increases PI3K/AKT/mTORC1 activity in ER-positive breast cancer cells [16]. As first shown by Bosch et al., inhibition of the PI3K pathway in ER-positive breast cancer results in induction of ER-dependent transcriptional activity. This in turn leads to an increase in ER signalling and increased ER dependence in ER-positive breast cancer [17]. These effects are suppressed by the addition of fulvestrant or tamoxifen. Recent studies have shown that after treatment with alpelisib and an aromatase inhibitor, *ESR1* mutations increase in number and allele fraction, which is associated with resistance [66] and provides a rationale for combination studies with novel selective oestrogen receptor down-graders (SERDs) such as elacestrant.

Yang et al. demonstrated that ER also drives PI3K/AKT activation in response to mTORC1 inhibition with everolimus [82].

## 4. The Future of Targeted Inhibitors of the PI3K/AKT/mTOR Pathway

### 4.1. Dual Inhibitors and Vertical Pathway Inhibition

There has been significant research into vertical inhibition of the PI3K/AKT/mTOR pathway in attempt to increase efficacy and improve clinical outcomes. Dual PI3K/mTOR inhibitors may limit the repression of negative feedback loops, which occurs when either kinase is targeted separately. Though this has shown to be a highly effective strategy preclinically at limiting the inhibition of negative feedback loops, to date, many of these compounds have failed in clinical trials both in terms of clinical outcomes and toxicities [83,84,85]. Dactolisib is one such compound which exhibited satisfactory anticancer effects in preclinical studies [86,87,88] but led to significant treatment-related adverse events, in particular, nausea, vomiting, and diarrhoea, alongside limited efficacy in phase I/II trials [89].

With the aim of reducing toxicity, an alternative proposed method of pathway inhibition is intermittent therapy. Gedatolisib is an intravenous dual inhibitor that targets mTOR and all class I isoforms of PI3K, dosed on a weekly schedule. This has shown encouraging antitumour activity and an overall manageable side-effect profile in phase I trials [90]. The phase III VIKTORIA-1 trial (NCT05501886) is currently recruiting patients with ER-positive, HER2-negative advanced breast cancer, with or without a *PIK3CA* mutation, who have been previously treated with a CDK4/6 inhibitor in combination with an aromatase inhibitor [91].

### 4.2. Combination Therapy

As PI3K inhibition increases ER transcriptional activity, this may be mitigated in part by CDK4/6 inhibition. CDK4/6 inhibition results in incomplete cell-cycle arrest and potentially early treatment resistance; therefore, combination with PI3K inhibition could lead to a more profound cell-cycle arrest [92]. Multiple preclinical studies have demonstrated synergy between CDK4/6 inhibitors and PI3K inhibitors, with PI3K inhibitors blocking the development of CDK4/6 inhibitor resistance in ER-positive, HER2-negative breast cancer [92,93]. The current first-line treatment for patients with metastatic ER-positive, HER2-negative breast cancer consists of a CDK4/6 inhibitor plus anti-hormonal therapy with an aromatase inhibitor or fulvestrant, depending upon prior endocrine therapy [94]. The majority of patients will eventually develop disease progression and a smaller subset of patients who are intrinsically resistant to CDK4/6 inhibitor/endocrine therapy combinations, who are not currently readily identifiable. Pascual et al. reported the first phase 1b study demonstrating safety and efficacy of the triplet combination of palbociclib, taselisib (a beta sparing PI3K inhibitor which is no longer under development), and fulvestrant in *PIK3CA*-mutant ER-positive, HER2-negative breast cancer, with a response rate of 37.5% (95% CI, 18.8–59.4) [95].

The phase III INAVO120 trial compared the combination palbociclib and fulvestrant with placebo versus inavolisib, an alpha-selective PI3K-specific inhibitor, as first-line therapy for patients with ER-positive, HER2-negative, *PIK3CA*-mutant advanced breast cancer who recurred on or within 12 months of adjuvant endocrine therapy. At the first interim analysis which was presented at the 2023 San Antonio Breast Cancer Symposium, inavolisib more than doubled PFS compared to placebo, 15.0 vs. 7.3 months (HR, 0.43; 95% CI, 0.32–0.59; *p* < 0.0001). Reassuringly, the toxicity profile of inavolisib was manageable, with a discontinuation rate of 6·8% due to toxicities (which included higher rates of mucositis, hyperglycaemia, and diarrhoea) reported [55]. At the time of writing, this trial has not been published, and the drug has not been approved by the regulatory authorities. In addition to INAVO120, inavolisib is being investigated in the phase III INAVO121 and INAVO122 studies in various combinations for patients with *PIK3CA*-mutant breast cancer. CAPItello-292 is another study which includes a triplet combination with a CDK4/6 inhibitor. This phase Ib/III study evaluates the safety and efficacy of capivasertib in combination with palbociclib and fulvestrant in patients with ER-positive, HER2-negative advanced breast cancer (NCT04862663). An important question that remains unanswered is whether all patients require triplet therapy and the additional toxicities of a combination approach. Early changes in ctDNA levels are predictive of PFS in advanced ER-positive breast cancer, with lack of ctDNA suppression correlating with shorter median PFS in an analysis of the PALOMA-3 trial of fulvestrant and palbociclib [96]. The open-label FAIM study (NCT04920708) is recruiting patients with advanced ER-positive, HER2-negative breast cancer on first-line fulvestrant and CDK4/6 inhibitor using ctDNA dynamic monitoring. Patients without ctDNA suppression at day 15 are then randomised to fulvestrant plus palbociclib with or without the AKT inhibitor, ipatasertib.

Up to 40% of patients develop an *ESR1* mutation after treatment with an aromatase inhibitor which contributes to endocrine resistance and potential increased sensitivity to SERDs [97]. Elacestrant and camizestrant are novel oral SERDs which have shown improved PFS when compared to fulvestrant in patients with *ESR1* mutations in the EMERALD and SERENA-2 studies, respectively. Elacestrant was approved by the FDA in 2023 for patients with ER-positive, HER2-negative advanced breast cancer and *ESR1* mutations following progression on at least one line of endocrine therapy. Merlino et al. evaluated the activity of the elacestrant in combination with the σ-sparing PI3K inhibitor MEN1611 in vitro and in vivo in breast cancer Patient-Derived Xenograft (PDX) models, resistant to CDK4/6 inhibitor and endocrine therapy, with mutations in *PIK3CA* and/or *ESR1*. In all the tested in vivo models, the combination of elacestrant and MEN1611 was superior in comparison to the single agents by overcoming resistance to ER inhibition [98]. This interesting strategy of combining PI3K inhibitors and SERDs needs further evaluation in patients with *ESR1* mutations before incorporation into clinical practice.

As discussed, the efficacy of mTOR inhibitors might be reduced by AKT activation due to IFG1R signalling. Xentuzumab is a humanised monoclonal antibody that binds to the IGF-1 and IGF-2 ligands and has demonstrated acceptable antitumour activity in Phase I studies, which lead to the phase II XENERA-1 trial [80,99]. This study evaluated the addition of xentuzumab to everolimus and exemestane in patients with ER-positive, HER2-negative advanced breast cancer. Though this combination could be safely delivered, no PFS benefit was seen with the addition of xentuzumab: median PFS 12.7 months with xentuzumab and 11.0 months with placebo (HR 1.19; 95% CI 0.55–2.59; *p* = 0.6534). This may be in part due to the fact that other pathways are able to upregulate on IGF inhibition. It is important to investigate whether predictive biomarkers establish who will derive clinical benefit from the combined inhibition of IGF1R and the PI3K/AKT/mTOR pathway.

Concomitant inhibition of the PI3K/AKT/mTOR and MEK-ERK has been shown to be effective preclinically [100], although clinical trials of these combinations have been discouraging, primarily due to toxicity preventing full suppression of both pathways. Both pathways are involved in a myriad of physiological functions which leads to significant challenges when managing the therapeutic indexes of these drugs combined. Mutant-selective PI3K inhibitors may help mitigate this issue. The combination of a mutant selective PI3K inhibitor and MEK/ERK pathway inhibitor could theoretically inhibit both PI3K and MEK/ERK pathways in cancer cells but only suppress MEK/ERK systemically, leading to reduced toxicity.

See Table 2 and Table 3 for recently accrued/currently recruiting trials for agents targeting the PI3K/AKT/mTOR pathway.

## 5. Conclusions

Considerable progress has been made over the past decade in the pharmacological targeting of the PI3K/AKT/mTOR pathway; however, the overall success of these compounds has been limited by intrinsic and acquired resistance which can occur through multiple mechanisms as discussed in this article. Another main cause for failure of these drugs in the clinical setting, which is not discussed in detail within the scope of this article, is the insufficient target inhibition at tolerated doses due to the narrow therapeutic index of these compounds, due to the critical role of this pathway in numerous physiological functions including glucose regulation. In this regard, the development of mutation-specific inhibitors and the innovative scheduling of treatments may help abrogate this.

The future potential for compounds that target the PI3K/AKT/mTOR pathway will depend on strategies involving combination treatments which mitigate the reactivation of the pathway, and other signalling pathways, that occurs via downregulation of feedback inhibition. This is an opportunity to design rationale-based combination therapies to increase the efficacy of PI3K inhibitors, as seen in INAVO120, adding the PIK3CA inhibitor to the standard combination of fulvestrant and palbociclib. Nonetheless, there is clearly much to learn about the optimal management of this pathway, and continued efforts to investigate novel strategies to overcome resistance and reduce toxicities will be extremely important to increase the practical efficacy of these targeted agents. Our review highlights the need for further research into the mechanism of resistance to inhibitors of this key mitogenic pathway, in particular, the newly approved AKT1 inhibitor, capivasertib.

## Figures and Tables

**Figure 1 cancers-16-02259-f001:**
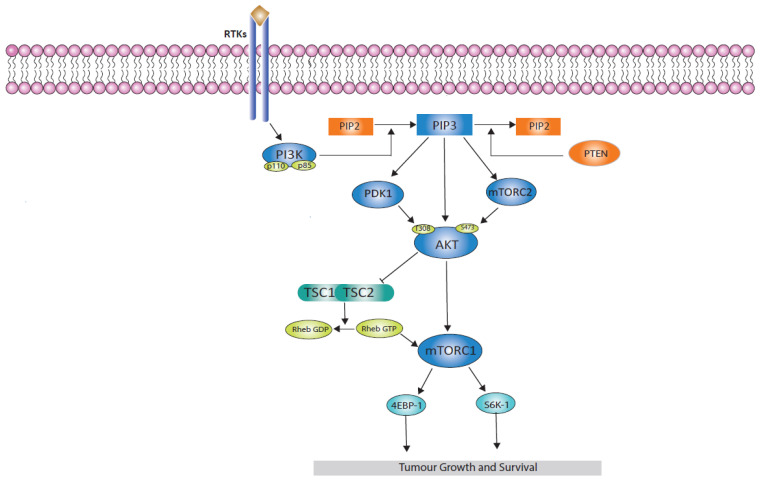
PI3K/AKT/mTOR pathway in breast cancer illustrating activation of PI3K leading to a signalling cascade including AKT and mTORC1 activation and resulting in tumour growth and survival.

**Figure 2 cancers-16-02259-f002:**
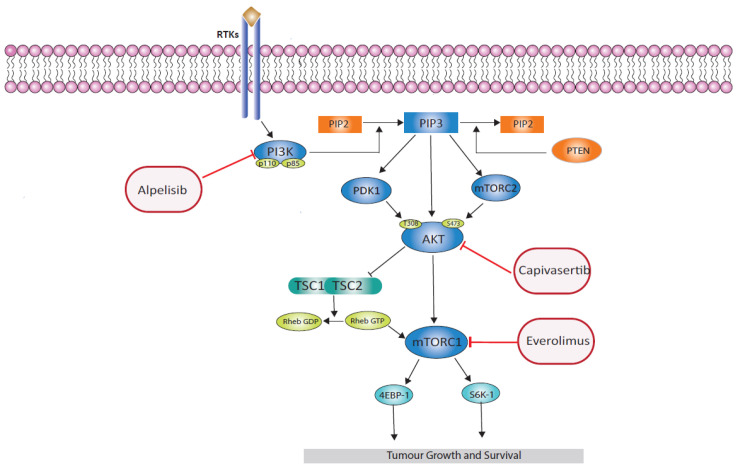
PI3K/AKT/mTOR pathway including an illustration of how the approved targeted inhibitors in ER-positive, HER-negative advanced breast cancer block the signalling cascade, thereby inhibiting tumour growth.

**Figure 3 cancers-16-02259-f003:**
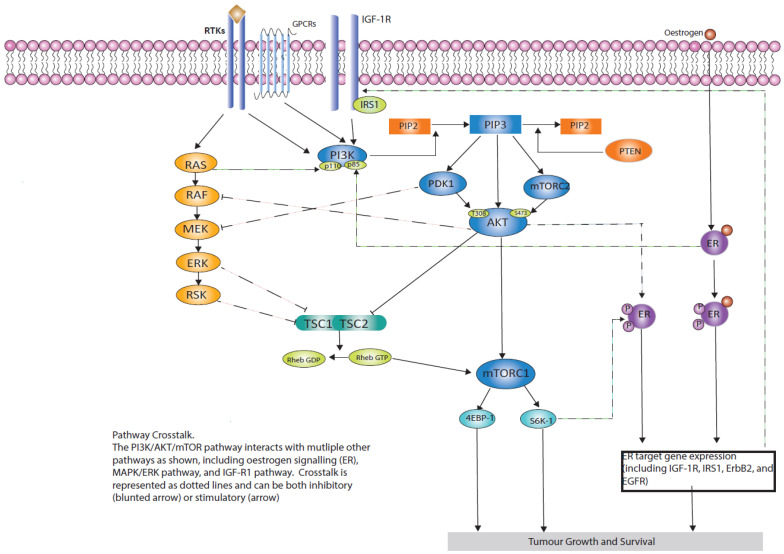
PI3K/AKT/mTOR pathway crosstalk with multiple pathways including ER signalling, the MAPK/ERK pathway and IGFR1 pathway. Crosstalk may be inhibitory or stimulatory.

**Table 1 cancers-16-02259-t001:** Principal phase 2/3 trials of agents targeting the PI3K/AKT/mTOR pathway combined with endocrine therapy in advanced ER-positive/HER2-negative breast cancer.

Drug Class	Drug Name	Trial Name and Sample Size	Treatment Arms	Response Rate (%)	Median PFS (Months)	Median OS (Months)
mTOR inhibitors	Everolimus	BOLERO-2 [28,29] *n* = 724	Exemestane + placebo	7.0 ^a^	2.8 ^a^	26.6 ^a^
			Exemestane + everolimus	0.4 ^a^	6.9 ^a^	31.0 ^a^
	Vistusertib	MANTA [36] *n* = 333	Fulvestrant	25.0 ^a^	5.4 ^a^	24.2 ^a^
			Fulvestrant + everolimus	41.2 ^a^	12.3 ^a,e^	NR ^a^
			Fulvestrant + daily vistusertib	30.4 ^a^	7.6 ^a^	27.1 ^a^
			Fulvestrant + intermittent vistusertib	28.6 ^a^	8.0 ^a^	24.2 ^a^
PI3K inhibitors	Alpelisib	SOLAR-1 [43] *n* = 572	Fulvestrant + placebo	12.8 ^b^	5.7 ^b^	31.4 ^b^
			Fulvestrant + alpelisib	26.6 ^b^	11.0 ^b^	39.3 ^b^
		BYLieve cohort A [45] *n* = 127	Fulvestrant + alpelisib	19.0 ^b^	8.0 ^b^	27.3 ^b^
	Inavolisib	INAVO-120 [55] *n* = 325	Fulvestrant + palbociclib + placebo	25.0 ^b^	21.1 ^b^	31.1 ^b^
			Fulvestrant + palbociclib + inavolisib	58.4 ^b^	46.2 ^b^	NR
AKT inhibitors	Capivasertib	CAPItello-291 [52] *n* = 708	Fulvestrant + placebo	12.2 ^a^	3.6 ^a^	nr*
				9.7 ^c^	3.1 ^c^	nr*
			Fulvestrant + capivasertib	22.9 ^a^	7.2 ^a^	nr*
				28.8 ^c^	7.3 ^c^	nr*
		FAKTION [53] *n* = 140	Fulvestrant + placebo	8.0 ^a^	4.8 ^a^	23.4 ^a^
				11 ^d^	4.6 ^c^	20 ^c^
			Fulvestrant + capivasertib	29.0 ^a^	10.3 ^a^	29.3 ^a^
				47 ^d^	12.8 ^c^	38.9 ^c^

PFS, progression free survival; OS, overall survival; NR, not reached; nr*, not reported; ^a^, allcomers irrespective of mutation status; ^b^, patients with *PIK3CA* mutation; ^c^, expanded biomarker analysis: patients with *PIK3CA*/*AKT1*/*PTEN*-alterations; ^d,^ patients with *PIK3CA*/*PTEN* alterations; ^e^, everolimus + exemestane is licenced indication, everolimus + fulvestrant is not licenced.

**Table 2 cancers-16-02259-t002:** Current phase 2/3 trials for agents that target PI3K/AKT/mTOR pathway in advanced ER-positive/HER2-negative breast cancer.

Drug	Clinical Trials.gov Identifier; Trial Name	Combined Therapy	Phase	Status	Planned (n)
Alpelisib	NCT05038735 (EPIK-B5)	Alpelisib + fulvestrant	3	Open (Recruiting)	234
	NCT04762979	Alpelisib + continued ET	2	Open (Recruiting)	44
Inavolisib	NCT05646862 (INAVO121)	Inavolisib + fulvestrant vs. alpelisib + fulvestrant	3	Open (Recruiting)	400
	NCT03424005 (Morpheus-panBC)	Multiple treatment arms	1/2	Open (Recruiting)	242
Capivasertib	NCT04862663 (CAPItello-292)	Capivasertib + CDK4/6i + fulvestrant	1b/3	Open (Recruiting)	850
	NCT05720260	Capivasertib + goserelin + fulvestrant +/− durvalumab	2	Open (Recruiting)	56
	NCT05563220 (ELEVATE)	Arm E: Capivasertib + elacestrant	1b/2	Open (Recruiting)	60
Ipatasertib	NCT04650581 (FINER)	Ipatasertib + fulvestrant	3	Open (Recruiting)	250
	NCT04920708 (FAIM)	Ipatasertib + palbociclib + fulvestrant	2	Open (Recruiting)	174

ET, endocrine therapy; CDK4/6i, cyclin dependent kinase 4 and 6 inhibitor.

**Table 3 cancers-16-02259-t003:** Current trials for novel dual inhibitors and mutant selective inhibitors that target PI3K/AKT/mTOR pathway in advanced ER-positive/HER2-negative breast cancer.

Drug Class	Target	Clinical Trials.gov Identifier; Trial Name	Drug	Combination or Monotherapy	Phase	Status	Planned (n)
Dual inhibitors	Class I PI3K + mTOR	NCT05501886 (VIKTORIA-1)	Gedatolisib	Multiple treatment arms	3	Open (recruiting)	701
Mutant selective PI3K inhibitors	Mutant selective H1047R	NCT05307705 (PIKASSO-01)	LOXO-783	Multiple treatment arms	1	Open (recruiting)	400
	Mutant selective H1047X	NCT05768139 (SCORPION)	STK-478	STX-478 + Fulvestrant	1/2	Open (recruiting)	220
	Mutant selective PI3Kα	NCT05216432 (ReDiscover)	RLY-2608	RLY-2608 + Fulvestrant	1/2	Open (recruiting)	400

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
