# Peer review of "Resistance to Targeted Inhibitors of the PI3K/AKT/mTOR Pathway in Advanced Oestrogen-Receptor-Positive Breast Cancer"

_cancers, 2024, doi:10.3390/cancers16122259_

Round 1

Reviewer 1 Report (Previous Reviewer 1)

Comments and Suggestions for Authors

The authors have significantly improved the manuscript. However, some references are very old and should be updated with the latest literature.

Author Response

This manuscript is a resubmission of an earlier submission. The following is a list of the peer review reports and author responses from that submission.

Round 1

Reviewer 1 Report

Comments and Suggestions for Authors

The paper authored by Iseult M. Browne and Alicia F. C. Okines demonstrates a commendable level of structure and informativeness. However, prior to the publication of the manuscript, several suggestions are provided below for the authors to consider addressing:

  1. What is the significance of the current study.
  2. The author made a formatting error throughout the manuscript by placing the full stop (.) before the numbered references instead of after the reference. This error needs to be corrected throughout the entire manuscript to ensure consistency and adherence to proper citation formatting standards.
  3. The abstract and introduction of the manuscript lack information about breast cancer, including general information and statistics. It is crucial to incorporate relevant data on breast cancer, such as prevalence, incidence rates, risk factors, and the significance of the topic in public health. By adding this information, the manuscript will provide a more comprehensive overview of the subject matter and highlight the importance of the research within the context of breast cancer research.

Authors can follow the pattern written in https://www.mdpi.com/1422-0067/22/22/12455

  1. References 1 and 2 is very old. Should be updated by inserting newer:

 https://www.mdpi.com/2073-4409/11/14/2209

  1. Figures legends are missing in Figure 1. PI3K/AKT//mTOR pathway, need to be written by the authors.
  2. The figure legends for Figure 2 are also incomplete. It needs to fix them.
  3. Add a "Reference" column to Table 1 and include proper references for each inhibitor.
  4. Please redraw Figure 3 using BioRender to address the issue of blurred arrows and ensure that the column for ER target genes is corrected. Ensure that the legends clearly describe the content of the figure.
  5. Add a "Reference" column to Table 2 & 3 also and include proper references for each inhibitor.
  6. The conclusion does not sufficiently incorporate the information. Additional details should be included to enhance clarity and precision.
  7. Minor English language correction is required to improve the quality of the review.
  8. Many references are very old which should be updated by inserting the latest updates.

By addressing these suggestions, the authors can further enhance the quality and impact of their manuscript before its publication.

Comments on the Quality of English Language
  1. Minor English language correction is required to improve the quality of the review.

Reviewer 2 Report

Comments and Suggestions for Authors

The manuscript entitled, “Resistance to Targeted Inhibitors of the PI3K/AKT/mTOR Pathway in Advanced Oestrogen Receptor-Positive Breast Cancer”, reviews signal transduction pathways, resistance mechanisms, and therapy development of an important subset of advanced breast cancer. The authors provide appropriate, quality content and commentary on the topic. Overall, the manuscript is well-written and sufficient with the exception of some minor critiques listed below.

·       The manuscript would benefit from reorganization and restructuring to improve clarity and eliminate repetition in the text. The authors give brief introductions to each topic early in the manuscript, but later repeat introductions when discussing each topic in depth. This disrupts the logical flow of the text.

·       The authors should incorporate more effective, non-redundant graphics. Information provided in figures 1, 2, and 3 overlap heavily. Resistance mechanisms and negative feedback are discussed in the text, but the information is complex and a figure would greatly aid the reader.

·       In figure 3 the lines are very thin and may be lost if viewed at low resolution. Biorender is a software tool that could help with improving figures.

·       The authors should expand upon specific sections of the text. The level of detail given for PI3K inhibitors is excellent, and the level of detail for mTOR and Akt inhibitors should match. Additionally, the authors should clarify when a mechanism is definitively known and when it is still in need of further scientific research.

·       Via a search of the ClinicalTrials.gov database we identified the clinical trial SABINA (identifier NCT05810870), which is testing MEN1611 monotherapy or in combination with Eribulin. This trail may be of interest to the authors for inclusion in this manuscript.

Round 2

Reviewer 1 Report

Comments and Suggestions for Authors

Although authors tried to improve the manuscript somewhat, but they did not take the reviewer's opinion seriously. Therefore, in its current format, I cannot endorse the manuscript for publication.

Comments on the Quality of English Language

Minor English corrections are required.
